# Structural Polymorphisms of Chromosome 3Aᵐ Containing *Lr63* Leaf Rust Resistance Loci Reflect the Geographical Distribution of *Triticum monococcum* L. and Related Diploid Wheats

Aleksandra Noweiska, Roksana Bobrowska and Michał Tomasz Kwiatek *

Department of Genetics and Plant Breeding, Faculty of Agronomy, Horticulture and Bioengineering, Poznań University of Life Sciences, 11 Dojazd Str., 60-632 Poznań, Poland; aleksandra.noweiska@up.poznan.pl (A.N.); roksana.bobrowska@up.poznan.pl (R.B.)
* Correspondence: michal.kwiatek@up.poznan.pl

**Abstract:** Wheat is one of the world's crucial staple food crops. In turn, einkorn wheat (*Triticum monococcum* L.) is considered a wild relative of wheat (*Triticum aestivum* L.) and can be used as a source of agronomically important genes for breeding purposes. Cultivated *T. monococcum* subsp. *monococcum* originated from *T. monococcum* subsp. *aegilopoides* (syn. *T. boeticum*). For the better utilization of valuable genes from these species, it is crucial to discern the genetic diversity at their cytological and molecular levels. Here, we used a fluorescence in situ hybridization toolbox and molecular markers linked to the leaf rust resistance gene *Lr63* (located on the short arm of the 3Aᵐ chromosome—3AᵐS) to track the polymorphisms between *T. monococcum* subsp. *monococcum*, *T. boeticum* and *T. urartu* (A-genome donor for hexaploid wheat) accessions, which were collected in different regions of Europe, Asia, and Africa. We distinguished three groups of accessions based on polymorphisms of cytomolecular and leaf rust resistance gene *Lr63* markers. We observed that the cultivated forms of *T. monococcum* revealed additional marker signals, which are characteristic for genomic alternations induced by the domestication process. Based on the structural analysis of the 3AᵐS chromosome arm, we concluded that the polymorphisms were induced by geographical dispersion and could be related to adaptation to local environmental conditions.

**Keywords:** chromosome alternations; diploid wheat; species dispersion; domestication; einkorn; leaf rust resistance

## 1. Introduction

Crop wild relatives (CWRs) are considered wild species that have sufficient levels of interfertility with other crops [1]. CWRs carry many beneficial traits for breeding, especially those, which have been lost during domestication and breeding selection, as well as novel adaptive alleles that can enhance crop diversity and productivity [2]. Initially, the classification for CWRs was established through the empirical crossing that resulted in four major germplasm categories: primary (no crossing barriers), secondary (benign crossing barriers), tertiary (requires special methods to obtain hybrid organisms, such as embryo rescue), and quaternary (genetic engineering technics are necessary to be performed) [1]. *Triticum monococcum* L. (2n = 2x = 14 chromosomes; AᵐAᵐ) is closely related to *Triticum urartu* Thumanjan ex Gandilyan (2n = 2x = 14; AᵘAᵘ), which has been reported as one of the primary gene pool ancestors of hexaploid wheat (*Triticum aestivum* L.; 2n = 6x = 42; BBAADD) from which the A-genome originated [3]. *T. monococcum* presents a high genetic variability, making it a significant gene pool for other species. Due to its close affinity to common wheat, it has been reported as a source of valuable genes, i.e., disease resistance genes, including leaf rust resistance genes [4]. Leaf rust (*Lr*) is a fungal disease caused by *Puccinia triticina* Eriksson. It is the most widespread of the wheat rust diseases, which

occurs in almost all growing areas and limits wheat production worldwide. The disease can take the form of an epidemic, which can lead to severe economic losses [5]. Until now, over 80 *Lr* resistance genes have been identified within the Triticeae tribe [6]. Some of these genes have been used to develop resistance to leaf rust in wheat varieties [5], including the *Lr63* gene, which is located on a short arm of chromosome 3A$^m$ (3A$^m$S) of *T. monococcum*. It is the only mapped leaf rust resistance gene on the distal part of chromosome 3A$^m$S, and is linked with microsatellites locus *Xbarc321* and *Xbarc57* markers (2.9 cM) [7].

Crop domestication is a process inseparably linked to the transition from hunter–gatherer societies to settled agriculture (the 'Neolithic revolution'; [8]), which independently appeared over a dozen times in various regions around the world from 10,000 to 12,000 years ago (ya) ca., to as recently as 3000–4000 ya [9–11]. This process can be called a conscious artificial selection of plants used in order to enhance their relevance to human demands, such as flavor, harvest, preservation, and methods of breeding. However, this process has been reported to be unconscious as well [12]. It is claimed that the factors which are responsible for early domestication include (1) the relocation of plants to new environmental niches, (2) human migration, and (3) genetic and genomic alternations, which are specific to selection [12]. Demographic effects often associated with domestication resulted in conspicuous impacts on genomic architecture, such as reductions in ineffective population sizes, reductions in diversity, and changes in the mating system, as well as targeted selection of specific traits [13,14]. There are also a number of reports showing evidence of large-scale chromosomal structural changes [15], changes in repetitive sequence content [16], and changes in gene variations and their copy numbers [17]. Cultivated *T. monococcum* L. subsp. *monococcum* originated from *T. boeticum* (syn. *T. monococcum* subsp. *aegilopoides*), which was widespread in southern Europe and western Asia. Even before domestication, *T. boeticum* was divided by the strong genetic differentiation into three races: race α, race β, and race γ [18]. However, only one race (β) has been exploited by mankind [18]. Precisely, wild *T. boeticum* was domesticated in the Karacadag mountains in southeast Turkey [19] and spread to several locations of the Fertile Crescent as the first cultivated wheat, called einkorn (*T. monococcum* L. subsp. *monococcum*) (Kilian et al. 2007). The common name was derived from the German "Einkorn", which means 'single grain', and relates to the occurrence of only a single grain per spikelet [20]. However, the name einkorn is used sometimes for both the wild (subsp. *aegilopoides*) and the cultivated (subsp. *monococcum*) forms. This cereal was important in the early Neolithic agriculture, but now is extensively grown in western Turkey, the Balkans, Switzerland, Germany, Spain, and the Caucasus [21]. During the last 5000 years, einkorn was eradicated and replaced by tetra- and hexaploid wheat. What is interesting is that both *T. monococcum* and *T. boeticum* (*T. monococcum* subsp. *aegilopoides*) are reproductively isolated from wild *T. urartu* (a progenitor of the A-genome of hexaploid wheat) with interspecific hybrids being sterile, although the two wild forms have comparable morphologies [22]. Cultivated einkorn is represented by a broad genetic variation [20]. This taxon includes nearly twenty identified botanical varieties [23] and six ecogeographical groups. The geographical diversification of *T. monococcum* from the domestication area was well restored through grain remains discovered in archaeological excavations [18,20], and is an example of well-documented speciation in the background of time and location.

What is interesting is that *T. monococcum*, *T. boeticum,* and *T. urartu* are karyotypically similar and have similar Giemsa C-banding patterns [24]. However, it was reported that fluorescence in situ hybridization (FISH), which allows for the direct localization of DNA sequences on chromosomes, revealed a number of intra- and interspecific divergence within diploid species of wheat [4,25–28]. FISH in plants commonly involves the application of probes containing conservative high-copy sequences. One of them is the *Afa*-family DNA probe, which is one of the most useful ones for the chromosome identification of diploid A-genome wheats. The *Afa*-family probe allowed for the recognition of the majority of chromosomes of cultivated einkorn [28] and *T. urartu* [4]. Other repeat DNA families, which were isolated from the bread wheat genome, among others, include pTa-86, pTa-465, pTa-

535, and pTa-713 [29–31] which can also provide informative labelling patterns for wheat chromosome discrimination. Among them, the probe pTa-535 was reported to generate the largest number of signals on the A-genome chromosomes, which are chromosome-specific. This clone is a 342 bp tandemly repeated DNA sequence, showing ~80% homology with the clone pTa-173, a member of the *Afa*-family [29].

In this study, we used molecular cytogenetics to analyze the karyotypes of *T. monococcum* accessions originating from different regions of eastern Europe, western Asia, and North Africa. Precisely, we analyzed the structural changes of the short arm of chromosome $3A^m$, which were probably induced by geographical dispersion and adaptation to local environmental conditions using cytomolecular tools and molecular markers linked to the *Lr63* locus.

## 2. Materials and Methods

### 2.1. Plant Material

Sixteen diploid wheat genotypes (Table 1) collected from different geographical regions were used in this study. One hexaploid wheat (*Triticum aestivum* L. subsp. *aestivum*) 'was the reference for the presence of locus *Lr63* (GSTR 444). Chinese Spring (CS) wheat was used as a standard control and for molecular probe generation as well. All accessions were provided by the National Small Grain Collection located at the Agricultural Research Station in Aberdeen, WA, USA.

**Table 1.** Origin and presence of *Xbarc321* and *Xbarc57* markers linked to *Lr63* locus in tested diploid wheat. "+"—presence of marker; "-"—absence of marker.

| No. | Plant ID | Cultivar | Species | Origin | *Xbarc321* | *Xbarc57* |
|-----|----------|----------|---------|--------|------------|-----------|
| 1. | GSTR 444 | Lr63 | *Triticum aestivum* subsp. *aestivum* | Canada | + | + |
| 2. | CLTR17667 | - | *Triticum urartu* | Turkey | - | + |
| 3. | PI428316 | G3220 | *Triticum urartu* | Iran | - | + |
| 4. | PI225164 | Kaploutras | *Triticum monococcum* subsp. *monococcum* | Greece | + | - |
| 5. | PI428011 | G3224 | *Triticum monococcum* subsp. *aegilopoides* | Azerbaijan | + | + |
| 6. | PI554513 | 84TK154-028.00 | *Triticum monococcum* subsp. *aegilopoides* | Soviet Union | + | - |
| 7. | PI668147 | Kromeriz | *Triticum monococcum* subsp. *monococcum* | Former Czechoslovakia | + | + |
| 8. | PI277130 | A TRI 613/59 | *Triticum monococcum* subsp. *monococcum* | Albania | + | + |
| 9. | PI614649 | UKR-99-075 | *Triticum monococcum* subsp. *aegilopoides* | Ukraine | + | - |
| 10. | PI290508 | V.J. 388 | *Triticum monococcum* subsp. *monococcum* | Hungary | + | + |
| 11. | PI662221 | GR05-052 | *Triticum monococcum* subsp. *aegilopoides* | Greece | + | - |
| 12. | PI307984 | K930 | *Triticum monococcum* subsp. *monococcum* | Morocco | - | + |
| 13. | CLTR17664 | - | *Triticum urartu* | Lebanon | - | - |
| 14. | PI170196 | 2498 | *Triticum monococcum* subsp. *monococcum* | Turkey | + | + |
| 15. | PI326317 | WIR 18140 | *Triticum monococcum* subsp. *monococcum* | Azerbaijan | + | + |
| 16. | PI591871 | SN-264 | *Triticum monococcum* subsp. *monococcum* | Georgia | + | + |
| 17. | PI487265 | SY 20033 | *Triticum urartu* | Syria | - | + |

### 2.2. Identification of Molecular Markers Linked to Lr63

GeneMATRIX Plant and Fungi DNA Purification Kit was used to perform DNA isolation from the leaves of 10-day-old seedlings (EURx Ltd., Gdansk, Poland). DNA quality and concentration were measured using a DeNovix spectrophotometer (DeNovix Inc., Wilmington, DE, USA) at the spectral length of 260 and 280 nm. The samples were diluted with Tris buffer (EURx Ltd., Gdansk, Poland) to attain a concentration of 50 ng/µL. To identify gene *Lr63*, the molecular markers *Xbarc57* and *Xbarc321* [7] were used. PCR reaction was carried out in 25 µL volumes, consisting of the following: 1µL of two primers (Sigma); 12.5 µL FastGene® Optima HotStart ReadyMix (NIPPON Genetics, Europe GmbH, Düren, Germany), which included FastGene® Optima DNA Polymerase blend (0.2 U per µL reaction), FastGene® Optima Buffer (1X), dNTPs (0.4 mM of each dNTP at 1X), MgCl$_2$ (4 mM at 1X), and stabilizers; 2 µL of DNA templates; and PCR-grade water. A PCR procedure was adjusted based on the standard protocol. The primer annealing temperatures of the marker primers were 52 °C for *Xbarc321* and 60 °C for *Xbarc57* [7]. The PCR final reaction included an initial denaturation at 94 °C for 5 min, followed by 35 cycles (denaturation, 94 °C for 45 s; primer annealing, 52 or 60 °C for 30s; elongation, 72 °C for 1 min), followed by the final extension for 7 min at 72 °C and storage at 4 °C. Polymerase chain reaction (PCR) was performed with the Labcycler thermocycler (SensoQuest GmbH, Göttingen, Germany). Amplification products were separated on 2% agarose gel (Bioshop, Canada Inc., Burlington, ON, Canada) in 1xTBE buffer (Bioshop, Canada Inc., Burlington, ON, Canada) for one and a half hours. Midori Green Advanced DNA Stain (Nippon Genetics Europe, Düren, Germany) was added to agarose gel. The UV Molecular Imager Gel Doc™ XR system with Biorad Bio Image™ software (Biorad, Berkeley, CA, USA) was used to visualize the amplification products.

### 2.3. Chromosome Preparation

Seeds were germinated in Petri dishes laid out with filter paper and flooded with water at room temperature (22 °C) for 4–6 days. After this time, root tips were cut off and stored in ice-cold water for 26 h. Fixation of the root tips was performed using ethanol and acetic acid (3:1, *v/v*). Mitotic preparations were created from root tips by digesting them with an enzyme mixture consisting of 20% (*v/v*) pectinase (Sigma), 1% (*w/v*) cellulose (Calbiochem), and 1% (*w/v*) cellulase 'Onozuka R-10' (Serva). By previously washing them with 0.01 M sodium citric buffer, slides were prepared according to the Kwiatek et al. [30] procedure.

### 2.4. DNA Molecular Probes

Genomic DNA from "CS" wheat was used to amplify the following repetitive sequences: pTa-86, pTa-535, and pTa-713, which were used as molecular probes (Table 2) [30]. According to Komuro et al. [30], two clones (pTa-535 and pTa-713) were determined to have especially valuable sequences for chromosome identification. In combination with pTa-86 (the pSc119 homologous sequence), these probes enabled the unambiguous discrimination of all wheat chromosomes, including orientation.

**Table 2.** Primer sequence and PCR terms for amplification of wheat repetitive sequences [29,30].

| Clone | NCBI GenBank Sequence Number | Primer Sequences (5′ to 3′) | Annealing Temperature (°C) |
|---|---|---|---|
| pTa-86 | KC290896 | ACGATTGACCAATCTCGGGG ACCGACCCAAATTACGAGAGT | 58.5 |
| pTa-535 | KC290894 | GCATAGCATGTGCGAAAGAG TCGTCCGAAACCCTGATAC | 59 |
| pTa-535 | KC290894 | GGGGCGGACGTCGTTG CCGTAAGATAGACAGGGTGGG | 59 |

The PCR mixture contained 12.5 μL of TaqNovaHS Master Mix, 1 μL of forward/reverse primers, 2 μL of DNA, and 8.5 μL of nuclease-free water. PCR reactions were performed under conditions of 95 °C for 3 min, 34 cycles of 95 °C for 30 s, annealing temperature at 59 °C for 30 s, 72 °C for 60 s, and 72 °C for 5 min. Labeling of molecular probes was performed using the nick translation method using the Nick Translation Kit (Roche/Merck). The pTa-535 sequence was labeled with tetramethyl-5-dUTP-rhodamine (Roche), whereas pTa-713 was labeled with digoxigenin-11-dUTP (Roche), and pTa-86 was labeled with Atto647 (Jena Bioscience, Jena, Germany).

### 2.5. DNA Molecular Probes

According to Kwiatek et al. [30], FISH was carried out with modifications. The repeat sequences of pTa-86, pTa-535, and pTa-713 were used as molecular probes. The hybridization mixture (10 μL/slide) contained: 50% formamide, 20% dextran sulphate, 10% 20×SSC, and 5% salmon sperm DNA and molecular probes. It was denatured at 70 °C for 10 min, 75 °C for 3 min, and then stored on ice for 10 min. Chromosomal DNA was denatured for 4 min at 70 °C with a hybridization mix and allowed to hybridize for 24 h at 37 °C. Digoxigenin-11-dUTP detection was conducted using antidigoxigenin-fluorescein antibody (Roche). Specific chromosomes were identified by comparing signal patterns and by comparing them to Komuro's et al. [29] work. Slides were analyzed at 1000× using a Delta Optical FMA050 microscope with a DLT-Cam PRO 12MP camera and DLT-Cam Viewer software. Image editing and karyotyping were performed using Adobe Photoshop C6 software.

## 3. Results

### 3.1. Intraspecific Polymorphism of Chromosome Markers

We performed a cytogenetic analysis based on the following probes: pTa-713, pTa-535, and pTa-86. All molecular probes provided signals on chromosomes of diploid wheat. FISH-painted chromosomes were categorized according to Komuro et al. [29]. The hybridization of the pTa-535 probe to chromosomes of einkorn species revealed clear and highly specific labeling patterns. All chromosomes carried 1–2 hybridization sites in chromosome-specific positions, although some intraspecific variation in signal localization and intensity has been observed. Hybridization patterns of pTa-86 in *T. urartu*, *T. monococcum* ssp. *Aegilopides*, and *T. monococcum* ssp. *Monococcum* were absent. In the *T. urartu* hybridization patterns of pTa-535, only two genotypes (CLTR17664 and PI487265) were observed on chromosome $3A^u$ in distal regions of long arms (Figure 1). The hybridization of the pTa-535 probe on the *T. monococcum* ssp. *Aegilopoides* chromosome revealed one very small signal in the subtelomeric region of the long arm in accession PI614649. A similar pattern was observed in the distal part of the short arm in genotype PI428011 of *T. boeticum*, and the additional hybridization sites of pTa-713 were found in the pericentromeric region (Figure 1).

Similar signal distributions of pTa-535 were observed in four accessions of *T. monococcum* ssp. *monococcum* (PI290508, PI591871, PI668147, and PI277130); one signal in the pericentromeric region of chromosome 3A in the hexaploid wheat overlap with sites of these accessions and the other pattern was noticed in the distal part of the short arms of chromosome $3A^{mm}$. Two accessions (PI170196 and PI326317) contained signals located in the subtelomeric region of the short arm. A weak signal was found in genotype PI225164 of einkorn in the subtelomeric region of chromosome $3A^{mm}L$ (Figure 1). In accessions CLTR17667, PI428316 (*T. urartu*); PI554513, PI662221 (*T. monococcum* ssp. *aegilopoides*), and PI307984 (*T. monococcum* ssp. *monococcum*), we observed a lack of hybridization patterns of pTa-535 and pTa-713 (Figure 1).

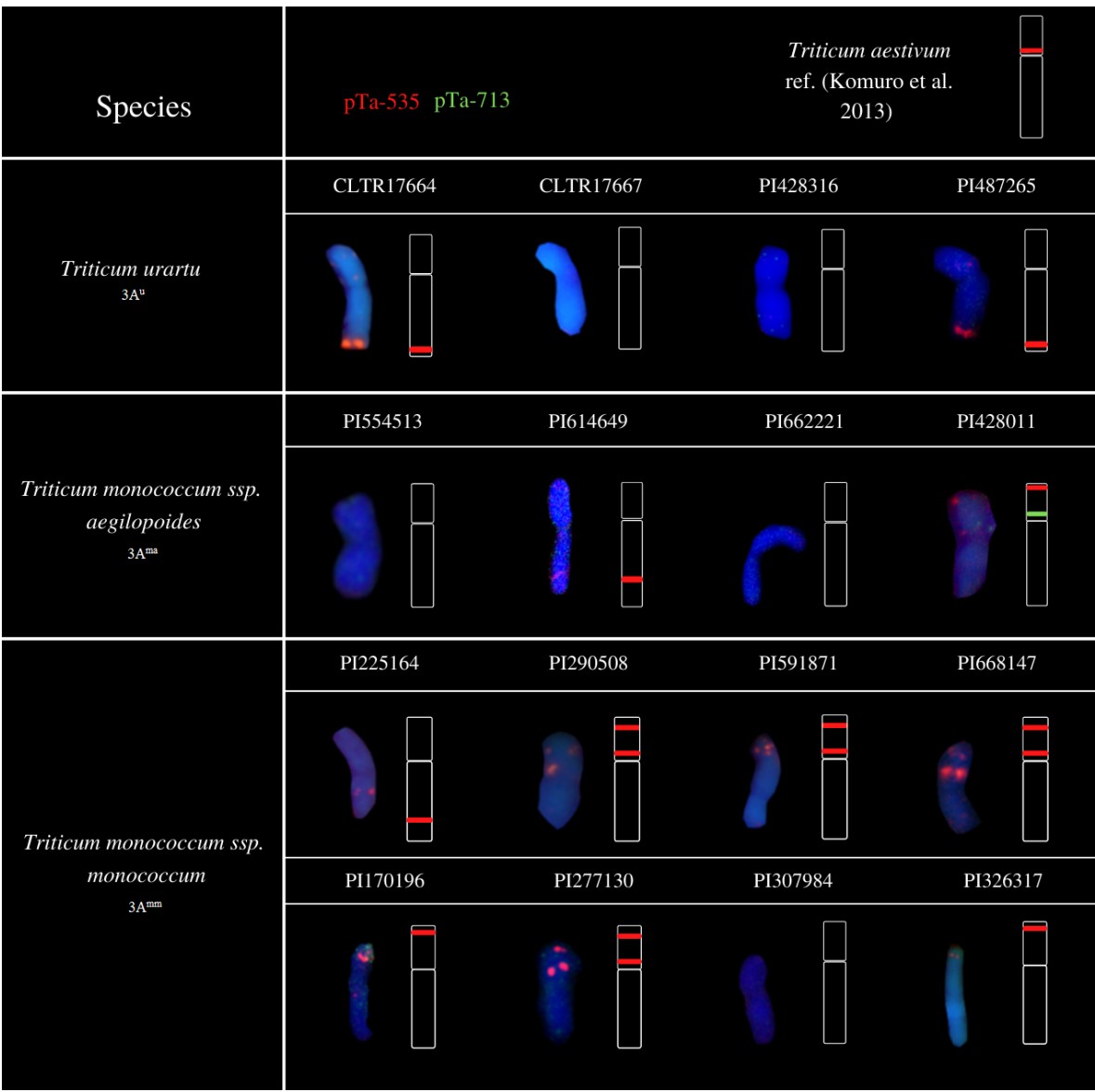

**Figure 1.** Chromosomes 3A after FISH with pTa-713 (**green**) and pTa-535 (**red**) probes of CLTR17664, CLTR17667, PI428316, PI487265, PI554513, PI614649, PI662221, PI428011, PI225164, PI290508, PI591871, PI668147, PI170196, PI277130, PI307984, and PI326317.

### 3.2. Polymorphism of Lr63 Loci

In parallel to the cytogenetic analysis, we analyzed the allelic variation in the *Lr63* leaf rust resistance loci at chromosome 3A (Figure 2). The expected specific product for marker *Xbarc321* was 191 bp according to Kolmer et al. [7]. In this experiment, 11 genotypes (PI 225164, PI 428011, PI 554513, PI 668147, PI 277130, PI 614649, PI 290508, PI 662221, PI 170196, PI 326317, and PI 591871) revealed PCR products, which were identical to one specific to GSTR 444 (reference genotype to locus *Lr63*) (Table 1 and Figure 2). Moreover, *Xbarc57* was used as the second marker in order to analyze the *Lr63* locus. Compared to the reference genotype (GSTR 444), the expected 240 bp products were identified in 11 genotypes (CLTR 17667, PI 428316, PI 428011, PI 668147, PI 277130, PI 290508, PI 307984, PI 170196, PI 326317, PI 591871, and PI 487265) (Table 1, Figure 2). The comparison of *Xbarc321* and *Xbarc57* marker analyses showed that both markers allowed to identify the *Lr63* gene locus in seven genotypes. Among them, six genotypes were considered as domesticated forms (*Triticum monococcum* subsp. *monococcum*) (PI 668147, PI 277130, PI 290508, PI 170196, PI 326317,

and PI 591871) and one was a nondomestication form (PI 428011) (*Triticum monocococcum* subsp. *aegilopoides*).

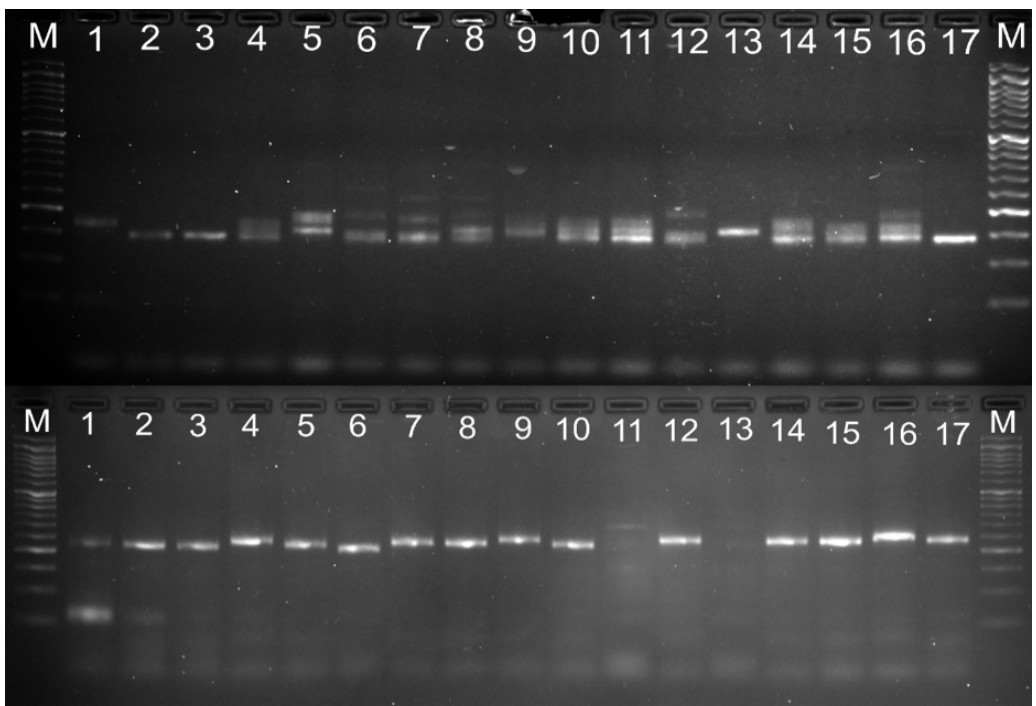

**Figure 2.** PCR amplification products of wheat genotypes with *xbarc321* and *xbarc57* markers linked to *Lr63* locus. M-50 bp DNA Ladder (NIPPON Genetics EUROPE GmbH); 1–17—wheat genotypes.

### 3.3. Rearrangements of Short Arm of 3A Chromosome

Taking into consideration the presence of molecular markers (*Xbarc321* and *Xbarc57*) and the signals of the pTa-535 probe on chromosome 3A of the short arm, it was possible to divide the accessions into three groups (Figure 3). The first group included seven genotypes carrying *Lr63* markers and possessed hybridization patterns on chromosome 3A. These accessions were as follows: PI 428011, PI 668147, PI 277130, PI 290508, PI 170196, PI 326317, and PI 591871, originating from Azerbaijan, Czechoslovakia, Albania, Hungary, Turkey, Azerbaijan, and Georgia (Figure 3). The genotypes in the second group were characterized by the presence of one marker (*Xbarc321* or *Xbarc57)* and the absence of the pTa-535 probe signal, including CLTR 17667, PI 428316, PI 225164, PI 554513, PI 614649, PI 662221, PI 307984, and PI 487265, originating from Turkey, Iran, Greece, the Soviet Union, Ukraine, Greece, Morocco, and Syria (Figure 3). One accession (CLTR 17664 from Lebanon), included in the last group, revealed the absence of *Lr63* markers and hybridization signals (Figure 3).

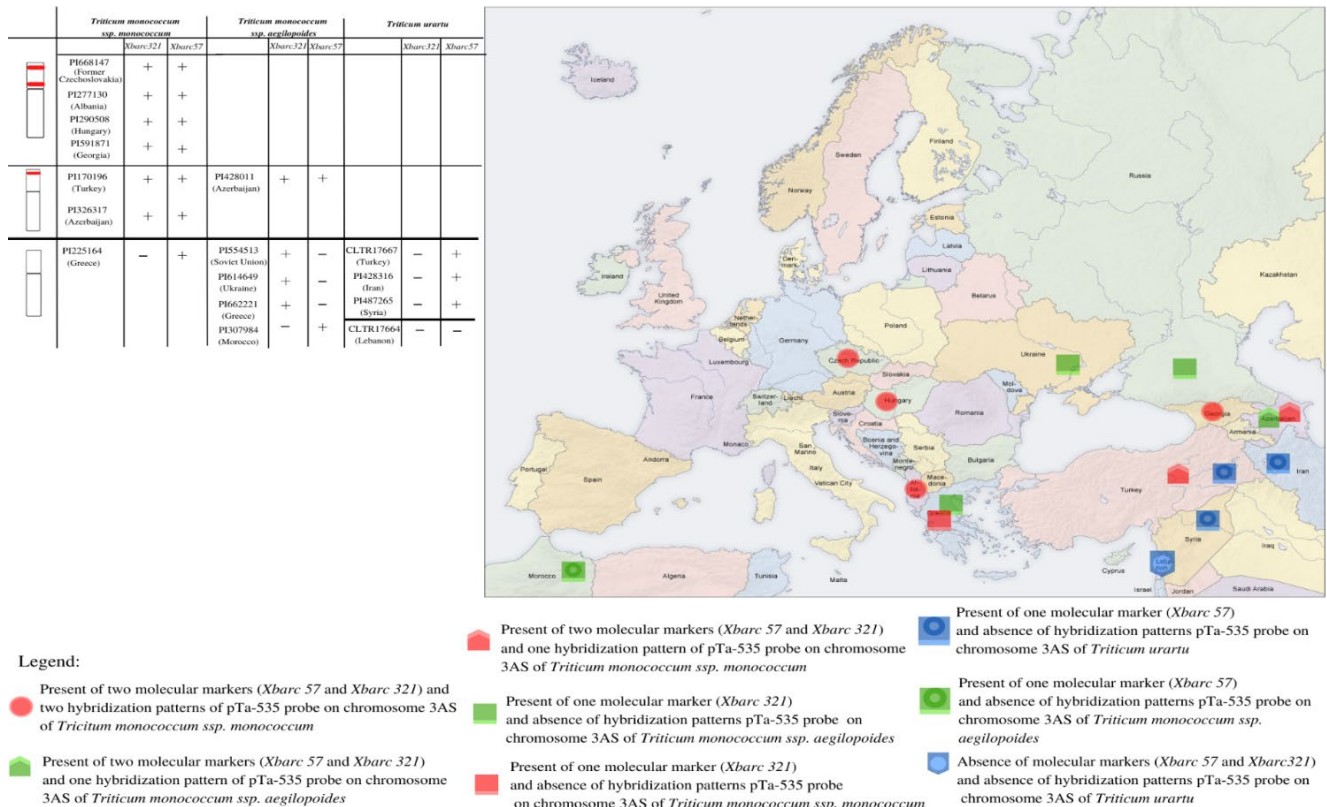

**Figure 3.** Geographical distribution of the studied accessions divided into three groups according pTa-535 signal location, as well as molecular markers, the presence of a particular amplification product.

## 4. Discussion

Revealing how domestication and selection impact disease-related genes is one of the crucial issues in plant genetics connected to resistance breeding. It has been reported that hexaploid wheat originated due to two hybridization events [32]. First, two wild species, *Triticum urartu* (A-genome donor) and an extinct species from the *Sitopsis* section (B-genome donor), hybridized and formed wild tetraploid wheat (*T. turgidum* ssp. *dicoccoides*). After the domestication of this wild form to cultivated tetraploid wheat (*T. turgidum* ssp. *dicoccum*), a second hybridization occurred with the wild grass *Ae. tauschii* (D-genome donor) resulting in the hexaploid wheat. The evolution of the resistance genes of wheat can be determined with comparative analyses and the allele mining of diverse germplasms. For example, the leaf rust resistance gene *Lr10* (located on chromosome arm 1AS) was cloned from bread cv. Thatcher *Lr10* [33]. It was reported that diploid (*T. urartu*; A-genome donor) and tetraploid (wild and domesticated) wheats carry a homologous sequence of *Lr10*, which has two haplotypes at the *Lr10* locus [34]. What is more, [35] suggested a balanced polymorphism and maintenance of both haplotypes of the *Lr10* gene sequence in the wheat gene pool, which is similar to the evolutionary pathway of genes in other species, such as *Rpm1* in Arabidopsis.

It is known that *T. monococcum* subsp. *aegilopoides*, *T. monococcum* subsp. *Monococcum*, and *T. urartu* carry the genome A, which is the axial subgenome for all wheats (*Triticum* sp.) [32]. Genes from these species can be introduced into wheat using direct crossing and chromosomal recombination. Diversified collections of these species are present in nature, widely distributed in different regions. The domestication of the diploid wheat *T. monococcum* ssp. *aegilopoides* was located in the geographical region of the Karaca Dağ (Karacadag) volcanic mountain, located in present-day south-eastern Turkey [18]. The wild form was first harvested and then transported to different geographical areas and cultivated there. Transport involved migrating farmers or exchanging seeds for other material goods,

because not all soils in the "Fertile Crescent" area were adjusted to cultivate crops. The directions of the early spread of diploid wheat included areas of present-day Turkey, Iraq, Syria, and Iran. In the later phase of agricultural expansion, crops were transported in an already nascent state of domestication [18,36]. According to Kolmer et al. [7], only one gene resistant to leaf rust (*Lr63*) has been mapped on the short arm of chromosome 3A$^m$. The short arm of the 3A$^m$ chromosome is also a region, where the main QTL for seed dormancy [37] *FLOWERING LOCUS T* (*FT*)-like, *TERMINAL FLOWER1* (*TFL1*)-like, and *MFT*-like [38] is located. Those loci are considered to be the most agronomically important traits considering crop domestication. It could, therefore, be stated that the selection pressure could alter this chromosome region with particular intensity. After all, it is known that in the transition from gathering to cultivation, early farmers selected CWRs with useful genetic modifications and developed improved populations with desirable traits [39]. The abovementioned traits, as well as the loss of seed dispersal mechanisms, increased grain size, the loss of vulnerability to environmental factors for germination and flowering, synchronous ripening, and a compact growth habit are included in the "domestication syndrome" characters, which are the most important adaptive traits selected by mankind [40]. In this research, we observed the presence of two markers (*Xbarc321* and *Xbarc57*) and hybridization patterns of probe pTa-535 on chromosome 3A accessions, which strongly indicated intra- and interspecific polymorphisms. The direction of the expansion of the first group of accessions proved the later phase of agricultural spread. The second group allowed to prove the early spread of einkorn, through directions of expansion and intra- and interspecific polymorphisms. The third group (*T. urartu*) was never domesticated and located near the Fertile Crescent. According to our cytogenetic observations, it could be stated that a similar organization of the 3A$^m$ chromosome (lack of pTa-535 signals and lack of one or two SSR markers) of *T. monococcum* spp. *monococcum* and *T. monococcum* spp. *aegilopoides* compared to *T. urartu* was observed only in accessions collected from the regions which were located closely to the domestication centers of einkorn wheat (Turkey, Azerbaijan, Greece, and Ukraine). Additional pTa-535 signals which appeared on the short arm of the 3A$^m$ chromosome were observed in the accessions, which revealed both markers linked to *Lr63* loci. Interestingly, those accessions were collected near the domestication center of *T. monococcum* (Turkey), as well as in central Europe (Hungary, former Czechoslovakia, and Albania), and Georgia. Hence, it could be stated that forms belonging to the first and second groups were more prone to be selected and perform desirable domestication traits. Both repetitive sequence redundancies and proximities to genes were reported to vary between cultivated and wild genotypes [41,42]. Such differences suggest the potential function of repetitive sequences in crop domestication. For example, differences in the proximity of retrotransposons to genes could contribute to the significant phenotypic differences between wild and cultivated sunflowers [43]. Recently, Ebrahizadegan et al. [44] reported that different classes of repetitive DNA sequences have differentially accumulated between *Aegilops tauschii* subsp. *strangulata* and the other two subspecies of *Ae. tauschii* that were in parallel with spike morphology, implying that factors affecting the so called "repeatome" evolution are variable even among highly closely related lineages. In our study, we observed that both markers linked with *Lr63* loci were present only in those accessions which revealed one or two additional pTa-535 sites. This chromosome organization pattern was characteristic to most of the *T. monococcum* subsp. *monococcum* accessions. Anker et al. [45] proposed that *Triticum monococcum* has a nonhost status to the pathogens responsible for leaf rust, and showed that *Triticum monococcum* subsp. *monococcum* is more rust-resistant compared to *Triticum monococcum* subp. *aegilopoides* and *Triticum urartu* [45].

## 5. Conclusions

Considering that new virulence pathotypes and races keep on appearing constantly, one of the key challenges for wheat breeders is the systematic development of new, elite varieties carrying effective resistance genes. In the description of the domestication model

of diploid wheat, Kilian et al. [36] reported that einkorn wheat retained a high level of genetic diversity in the domesticated lines, which can be used to improve common wheat. Alterations and reorganization of the repetitive DNA sequences are the strongest indications of evolution and speciation processes. Therefore, the identification of polymorphisms in chromosome structures of different accessions of einkorn wheat may be helpful in further basic and application research.

**Author Contributions:** M.T.K. initiated the project. A.N. conducted the experiments and analyses. A.N. and R.B. wrote the first draft of the manuscript. M.T.K. revised the draft and improved the manuscript. All authors have read and agreed to the published version of the manuscript.

**Funding:** This publication was co-financed by the framework of the Ministry of Science and Higher Education program as "Regional Initiative Excellence" in years 2019–2022, project No. 005/RID/2018/19.

**Informed Consent Statement:** Not applicable.

**Acknowledgments:** The authors would like to acknowledge Harrold Bockelman at the USDA/ARS Small Grains Laboratory, (Aberdeen, ID, USA), for providing the seed samples.

**Conflicts of Interest:** The authors declare no conflict of interest.

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
