# Peer review of "Structural Polymorphisms of Chromosome 3Am Containing Lr63 Leaf Rust Resistance Loci Reflect the Geographical Distribution of Triticum monococcum L. and Related Diploid Wheats"

_agriculture, doi:10.3390/agriculture12070966_

Round 1
Reviewer 1 Report
This research article entitled “Structural polymorphisms of chromosome 3A m containing Lr63 leaf rust resistance loci reflect the geographical distribution of Triticum monococcum L. and related diploid wheats” seems interesting to understand the impact of domestication and selection on disease tolerance. Overall the manuscript is well written and ease to understand that develop the reader interest. I provide some suggestions to improve the research article further for publication.
1. Figure 3 quality is not better, suggest to improve the figure quality.
2. Discussion part of the article needs to improve; develop the link between domestication and disease tolerance related genes (Lr63).
3. Discuss about how domestication and selection impact on disease related genes.
4. In the discussion section, some sentences have missing spaces that make confusion in reading.
Author Response
Many tanks for your constructive comments. Here are the responses for your questions:
- Figure 3 quality is not better, suggest to improve the figure quality.
The quality of the figure is adjusted to the journal’s requirements.
- Discussion part of the article needs to improve; develop the link between domestication and disease tolerance related genes (Lr63).
- Discuss about how domestication and selection impact on disease related genes.
We have improved and developed the discussion according to your comments.
- In the discussion section, some sentences have missing spaces that make confusion in reading.
We have edited the ms to remove the typos.
Reviewer 2 Report
it is recommended that all suggested points must be addressed by the author and updated references should by more in number and must be added both in introduction and discussion sections.

Author Response
- Add few lines about importance of wheat in abstract and introduction and also add objectives of the study in introduction.
The expanded information about the importance of wheat is presented in the introduction. We have inserted only one initial sentence to introduce the species.
2. What is room temp. in this case?
22 degrees of C (corrected i the text)
3. Please add reference for confirmation of mentioned wheat spp.
carrying A genome
Reference of Venske et al. 2019 was added
4. Add conclusion in this paper
and also mention utilization of this study in wheat breeding program
We have improved the conclusion chapter
5. Update references for recent year in introduction and discussion.
Done
Many thanks for your contribution and time!
Reviewer 3 Report
Authors described an association between structural polymorphisms of 3AmS introgression carrying Lr63 in 17 wheat genotypes and their geographical distributions. The overall findings may be interesting to certain wheat researchers, but the manuscript is not well-prepared. Here are some of my suggestions for the improvement of the manuscript:
Line 180: Why these three probes (pTa-713, pTa-535 and pTa-86), but not others, were selected? Illustrations in Line 97-103 required further explanations, particularly about pTa-86. More information about these probes should be introduced.
In Figure 1, I could not find any signals for pTa-86 as indicated in color of yellow. I did not find result for “GSTR 444”, which seemed to be an important control genotype carrying Lr63.
As a key gene used in this evolutionary study, functions of Lr63 in all the 17 genotypes should be tested. A leaf rust inoculation assay is needed to validate the existence of Lr63. PCR amplification using molecular markers could provide very few indirect evidence.
Lines 213-215 and 219-220: 11 or 17 genotypes? Statements should be consistent with Table 1 and Figure 2.
Figure 2 can be moved to supplementary materials since all the results in this low qualified figure have already been presented in Table 1.
Legend for Figure 3 was too simple. Consider revise it.
Discussion section: More paragraphs should be divided in Line 245-302. Other association studies on genetic loci and geographical dispersion should be referred and discussed. A conclusion of this study should be provided at the end of the discussion.
Author Response
1. Line 180: Why these three probes (pTa-713, pTa-535 and pTa-86), but not others, were selected? Illustrations in Line 97-103 required further explanations, particularly about pTa-86. More information about these probes should be introduced.
Two clones (pTa-535 and pTa-713) were determined to have especially valuable sequences for chromosome identification. In combination with pTa-86 (the pSc119 homologous sequence), these probes enable unambiguous discrimination of all wheat chromosomes including orientation.
2. In Figure 1, I could not find any signals for pTa-86 as indicated in color of yellow. I did not find result for “GSTR 444”, which seemed to be an important control genotype carrying Lr63.
We claimed in the results, that there were no pTa-86 signals observed. We have removed the description form the figure caption.
3. As a key gene used in this evolutionary study, functions of Lr63 in all the 17 genotypes should be tested. A leaf rust inoculation assay is needed to validate the existence of Lr63. PCR amplification using molecular markers could provide very few indirect evidence. PCR analyses may allow for initial selection; in later stages of the study, inoculations will be performed
The main assumption of this work was to identify the structural polymorphisms of chromosome 3Am containing Lr63 leaf rust resistance loci reflect the geographical distribution of Triticum monococcum L. and related diploid wheats . Our next goal is to check the level of the Lr63 expression. We are going to perform inoculation tests with different pathotypes and uredinia mixtures.
4. Lines 213-215 and 219-220: 11 or 17 genotypes? Statements should be consistent with Table 1 and Figure 2.
The 17 genotypes analysed are those described in the materials section of Table 1. The 11 genotypes are those whose product overlapped with the positive control genotype according to the electropherogram (Figure 2)
5. Legend for Figure 3 was too simple. Consider revise it.
The legend has been revised.
6. Discussion section: More paragraphs should be divided in Line 245-302. Other association studies on genetic loci and geographical dispersion should be referred and discussed. A conclusion of this study should be provided at the end of the discussion.
We have improved the discussion.
Round 2
Reviewer 3 Report
Authors have addressed most of my previous concerns.